# Silicate Melt Inclusions in Diamonds of Eclogite Paragenesis from Placers on the Northeastern Siberian Craton

**Vladislav Shatsky [1,2,*], Dmitry Zedgenizov [1,2], Alexey Ragozin [1,2] and Viktoriya Kalinina [1]**

[1]  V.S. Sobolev Institute of Geology and Mineralogy, Siberian Branch of Russian Academy of Science, 630090 Novosibirsk, Russia

[2]  Geology and Geophysics Department, Novosibirsk State University, 630090 Novosibirsk, Russia

\*   Correspondence: shatsky@igm.nsc.ru

**Abstract:** New findings of silicate-melt inclusions in two alluvial diamonds (from the Kholomolokh placer, northeastern Siberian Platform) are reported. Both diamonds exhibit a high degree of N aggregation state (60–70% B) suggesting their long residence in the mantle. Raman spectral analysis revealed that the composite inclusions consist of clinopyroxene and silicate glass. Hopper crystals of clinopyroxene were observed using scanning electron microscopy and energy-dispersive spectroscopic analyses; these are different in composition from the omphacite inclusions that co-exist in the same diamonds. The glasses in these inclusions contain relatively high $SiO_2$, $Al_2O_3$, $Na_2O$ and, $K_2O$. These composite inclusions are primary melt that partially crystallised at the cooling stage. Hopper crystals of clinopyroxene imply rapid cooling rates, likely related to the uplift of crystals in the kimberlite melt. The reconstructed composition of such primary melts suggests that they were formed as the product of metasomatised mantle. One of the most likely source of melts/fluids metasomatising the mantle could be a subducted slab.

**Keywords:** diamond; mantle; mineral inclusions; melt inclusions; diamond-forming fluids/melts

## 1. Introduction

Natural diamonds often contain mineral inclusions, and these inclusions can indicate the type of rock the host diamonds formed in [1–10]. At the same time, recent data from eclogite xenoliths have indicated the superimposed character of diamonds, suggesting that diamond crystallisation can result from metasomatism [11–21]. There have also been reports that single diamondiferous xenoliths can contain diamonds of several generations, suggesting multiple stages of diamond growth.

It has been found that microinclusions in fibrous and coated diamonds primarily represent diamond-generating high-density fluids (HDFs) [22–32]. Studies of these HDFs have revealed four compositional end members: (1) Saline HDFs that mostly contain K, Na, Cl and water, with some carbonates and silicates; (2) high-Mg HDFs, characterised by high MgO and carbonates, and low $SiO_2$, $Al_2O_3$ and water; and a continuous array between (3) silicic and (4) low-Mg HDFs, with varying abundances of silicates, low-Mg carbonates and water [23,26,33–36].

Weiss et al. [37] determined that fluid-rock interactions, combined with in-situ melting, could lead to compositional transitions in HDFs as saline fluids passed through the mixed peridotite-eclogite lithosphere. However, there have only been a few findings of HDF microinclusions in monocrystalline diamonds. Weiss et al. [38] reported on HDFs in monocrystalline diamonds from Finsch, South Africa and in the monocrystalline core of a coated diamond from Kankan, Guinea. Fluids compositionally similar to those occurring in fibrous diamonds have been found in twinned diamonds [39]. These suggest

that the majority of lithospheric diamonds were created from migrating carbonate-bearing HDFs that reacted with the host rocks at the base of the subcontinental lithosphere.

In addition to HDFs, there are several other indications that monocrystalline diamonds contain melt-derived inclusions. A polymineral aggregate was identified as an inclusion in a diamond from the Mir kimberlite pipe, Yakutia [40]. The melt inclusion was found to contain four phases—clinopyroxene, rutile, K-Al-Si and Fe-Ti-Si. The bulk composition of the inclusion was similar to that of hypersthene andesite [40]. Inclusions of highly-potassic phases, similar in composition to K-feldspar, were found in a coated diamond crystal from the Mir pipe, and were interpreted as melt inclusions [41]. In addition to these potassic phases, the host diamond also contained coesite and omphacite inclusions. It has been suggested that the transparent diamond core crystallised from the melt, while the oversaturated HDF phase represented the medium in which the diamond coating grew [41].

The amount of data available concerning the composition of diamond-forming environments does cover every possibility. Here, we report new findings of composite inclusions that contain silicate glass and hopper pyroxene crystals from within two alluvial diamonds (from the Kholomolokh placer, northeastern Siberian Platform; Figure 1). The diamond placer deposits of the northeastern region of the Siberian Platform are located in the Palaeoproterozoic Khapchan fold-belt of the Olenek tectonic province. The known kimberlites from this region are low-grade or non-diamondiferous. Diamonds from these placers have been classified into three groups: (1) typical octahedral to rounded crystals; (2) yellow-orange or dark grey cuboids; and (3) rounded, dark crystals, with radial-mosaic structure [42]. Diamonds of eclogitic paragenesis in these deposits are dominant, with abundances reaching 85% [43].

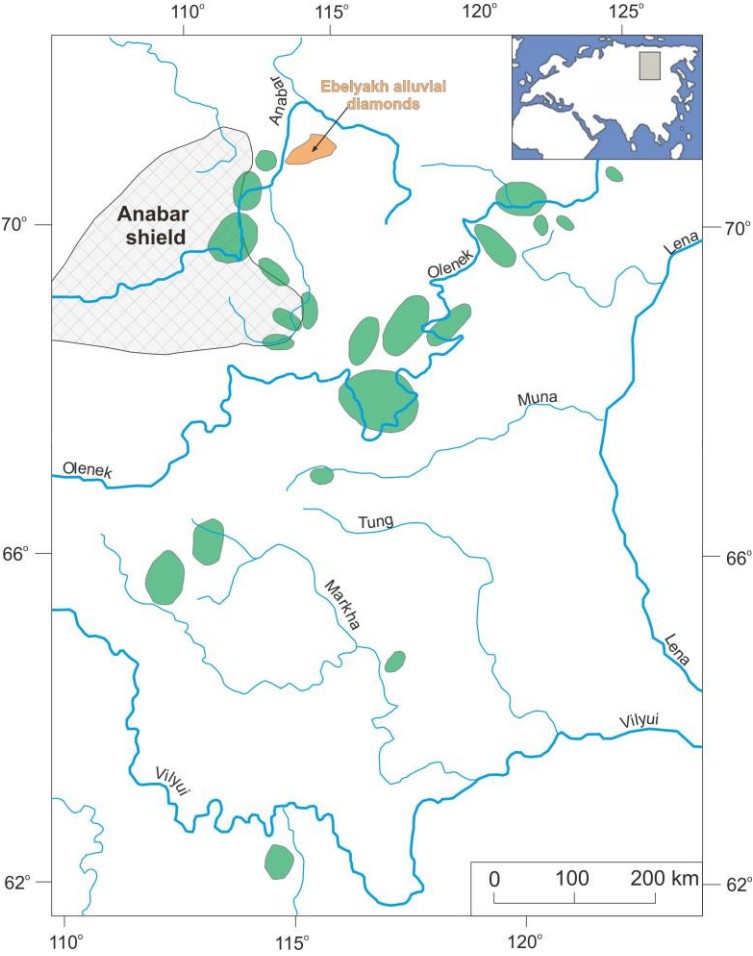

**Figure 1.** Location of Ebelyakh alluvial diamond deposits in the northeastern Siberian craton. The kimberlite fields are shown as green.

## 2. Samples and Methods

The two diamonds are elongated, rounded crystals with visible inclusions (Figure 2). The surfaces of both crystals indicate intensive resorption. The diamonds were polished on two sides, along two parallel dodecahedral planes, to expose their inclusions.

The samples were investigated at the Analytical Center for Multi-element and Isotope Research SB RAS (Novosibirsk, Russia). The infrared spectra of the diamonds were recorded using a Bruker VERTEX 70 Fourier-transform infrared (FTIR) spectroscope, equipped with a HYPERION 2000. Local spectra were measured over a range of 370–5000 $cm^{-1}$, with 30 scans at a resolution of 4 $cm^{-1}$. The spectra were recorded at an aperture of $60 \times 60$ μm. The contents of the N-centres were calculated following standard procedures. The intrinsic absorption of diamond (12.8 $cm^{-1}$ at 2030 $cm^{-1}$) was taken to be the internal standard [44]. The decomposition of the experimental lines between 1100 and 1350 $cm^{-1}$ made it possible to determine the contribution of different N-defects, using the characteristic lines of specified shapes. The contents of the N-defects were determined on the basis of proposed ratios [45,46]. The total uncertainty was better than 10% for N content, and the precision was approximately 5% (2σ) for the aggregation state. The results are summarised in Supplementary Table S1.

The major elements in the diamond inclusions were determined using a Camebax MICRO electron microprobe (EMP). The quantitative EMP analyses were performed at a 15 kV accelerating voltage, 20 nA sample current, and 2 μm beam diameter. The chemical compositions of the melt inclusions were examined by energy-dispersive spectroscopy (EDS), using a Tescan MIRA3 LMU scanning electron microscope (SEM) and AztecEnergy/X-max 50 EDS microanalysis software. A single set of reference samples was used, which included the simplest compounds and pure metals, including quartz/$SiO_2$ (for Si and O), corundum/$Al_2O_3$ (Al), $Cr_2O_3$ (Cr), blue diopside/$MgCaSi_2O_6$ (Mg and Ca), albite/$NaAlSi_3O_8$ (Na), orthoclase/$KAlSi_3O_8$ (K), and the metals Ti, Mn, Fe and Ni. The full protocol for the EDS measurements is provided in [47].

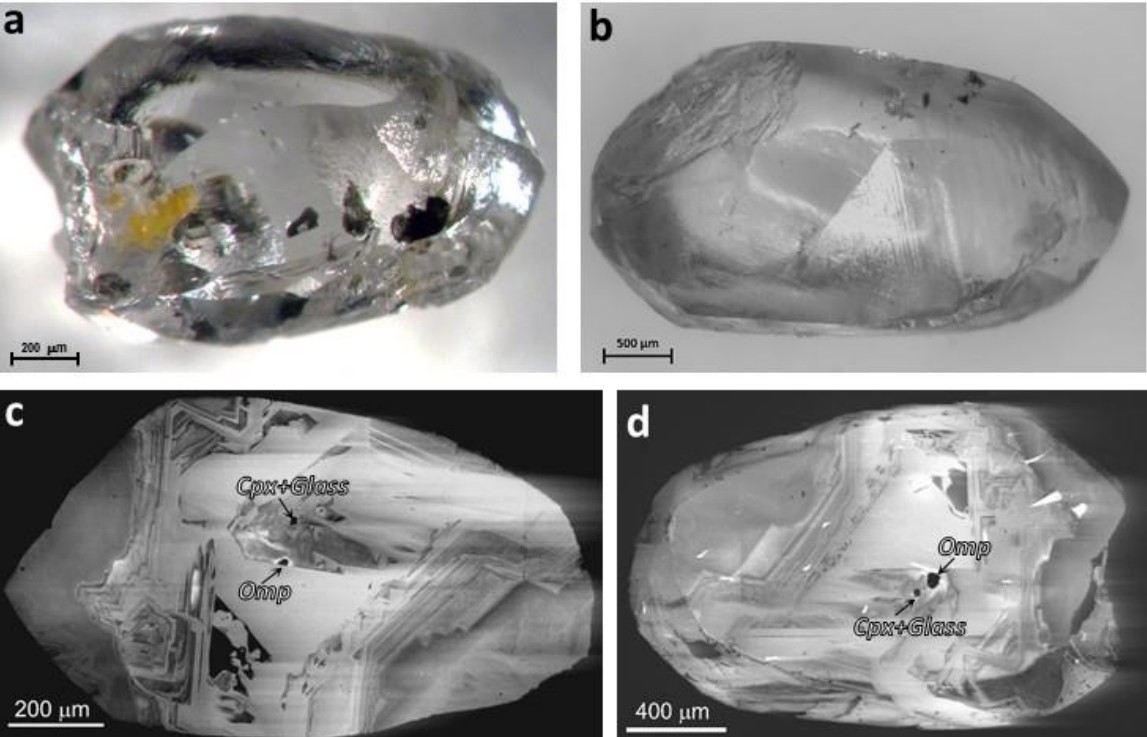

**Figure 2.** Microphotographs and cathodoluminescence images of polished plates from diamonds: (**a**,**c**)—Hi-39, (**b**,**d**)—Hi-51. Omp—omphacite inclusions, Cpx + Glass—combined clinopyroxene and glass inclusions.

Cathodoluminescence (CL) patterns from diamonds plates were studied using an Oxford Centaurus detector on a Leo-1430VP SEM (at accelerating voltages of 12–15 kV and with an electron beam current of ~0.5 mA).

The Raman spectral measurements were performed using a Horiba Jobin Yvon LabRAM HR800 spectrometer, equipped with an Olympus BX41 microscope. A 514 nm Ar-ion laser was used as an excitation source. The spectra were recorded at room temperature, in a backscattering geometry, with a laser power of about 1 mW and a spectral resolution of approximately 2 cm$^{-1}$.

## 3. Results

According to the FTIR spectroscopy, both diamonds exhibited a high degree of N aggregation (Supplementary Table S1). The content of the B1 centres varied between 61% and 69% in Sample Hi-51, while in Sample Hi-39, the %B was between 63% and 69%. The total N content was in the range of 324 to 596 ppm in Hi-51. There was no evidence for any regularity in the spatial N variability in this diamond. The highest N content in Hi-39 (541 ppm) was observed in the core and intermediate region of the crystal, while the N content was significantly lower (249 ppm) in the peripheral parts (Supplementary Table S1). However, the CL imaging of both crystals revealed a complicated history of growth and dissolution (Figure 1).

The Hi-51 diamond contained two adjacent inclusions (65 μm from each other)—omphacite and a composite inclusion. The latter consisted of two phases—melt glass (interpreted as quenched, non-recrystallised melt) and clinopyroxene. The clinopyroxene had a form that we interpreted as a hopper crystal (Figure 3a). The size of the inclusion was 50 μm. The Hi-39 diamond contained two omphacite inclusions and a composite, two-phase (glass and clinopyroxene) melt inclusion (Figure 3b). The diameter of the two-phase inclusion was 40 μm. The omphacite and two-phase inclusions in both diamonds were found at the centre. The omphacite inclusions had the same jadeite component (45%) and similar Mg# (74.5 in Hi-39 and 73.3 Hi-51; Table 1). The eskolaite content of the omphacites amounted to 13.4% in Hi 39, and 8% in Hi-51.

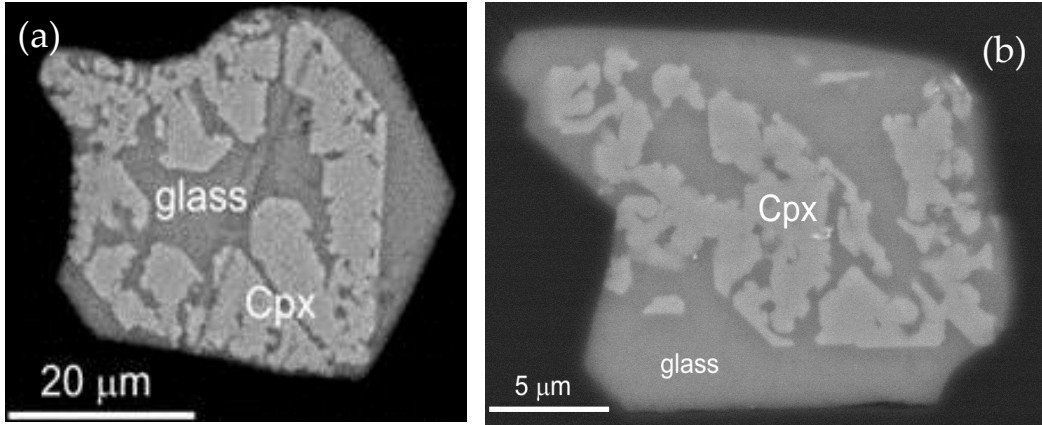

**Figure 3.** Back scattered electrons images of two-phase composite (melt) inclusion containing clinopyroxene (Cpx) and silicate glass in diamonds Hi-51 (**a**) and Hi-39 (**b**).

The Raman spectral analysis revealed that the composite two-phase inclusions included clinopyroxene (characteristic Raman shift at around 680 cm$^{-1}$ and 1015 cm$^{-1}$). The second phase most likely represents silicate glass, as the Raman spectra exhibited a strong, wide band at around ~890 cm$^{-1}$ and, in some cases, a weak, wide band at around ~380 cm$^{-1}$ (Figure 4).

The SEM and EDS observations showed that the hopper crystals in the composite two-phase inclusions were clinopyroxene, with compositions different from those of the omphacite inclusions (Table 1). The approximate compositions of the clinopyroxenes in these two-phase inclusions were quite similar (Table 1). The other phase, representing quenched, non-recrystallised melts, had low oxide

totals (Table 1). The composition of the glass resembles the composition of trachyte. The $Na_2O + K_2O$ content of this $SiO_2$-rich glass exceeded 12 wt %. Hi-39 contained no FeO, MgO or $TiO_2$, while the $SiO_2$ glass from the inclusions in Hi-51 was characterised by relatively high FeO, MgO, and $TiO_2$ contents (Table 1). Assuming that pyroxene occupies 50% of the mass of the inclusions in both diamonds, the reconstructed parental melt would be similar to basaltic trachyandesite, compositionally similar to those documented from eclogite partial-melting experiments [48,49].

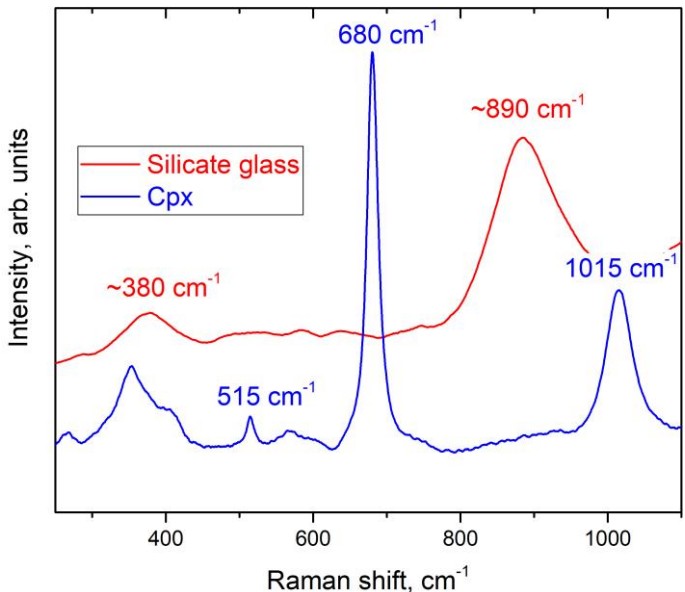

**Figure 4.** Raman spectra of composite two-phase inclusion from diamond Hi-39.

**Table 1.** Chemical compositions of inclusions in studied diamonds.

| Sample | Hi-39 | | | | Hi-51 | | | |
|---|---|---|---|---|---|---|---|---|
| Phase | Glass | Cpx | Rec. | Omp | Cpx | Glass | Rec. | Omp |
| $SiO_2$ | 59.6 | 49.2 | 54.5 | 55.4 | 51.1 | 61.2 | 55.8 | 54.9 |
| $TiO_2$ | | 2.78 | 1.39 | 0.880 | 1.08 | 0.80 | 1.01 | 0.82 |
| $Al_2O_3$ | 19.6 | 10.4 | 15 | 15.0 | 10.3 | 19.6 | 14.6 | 15.2 |
| $Cr_2O_3$ | | | | 0.04 | | | | |
| FeO | | 5.27 | 2.63 | 6.34 | 5.38 | 2.07 | 3.69 | 4.10 |
| MnO | | | | 0.06 | | | | |
| MgO | | 9.59 | 4.8 | 3.87 | 10.3 | 0.56 | 5.40 | 6.32 |
| CaO | 1.39 | 15.9 | 8.61 | 10.8 | 16.8 | 1.47 | 9.11 | 11.1 |
| $Na_2O$ | 7.40 | 3.99 | 5.7 | 6.59 | 3.72 | 5.67 | 5.46 | 6.51 |
| $K_2O$ | 4.65 | | 2.32 | 0.31 | | 5.57 | 2.67 | 0.48 |
| Total | 92.7 | 97.13 | 94.95 | 99.3 | 98.6 | 96.9 | 97.7 | 99.5 |
| Mg# | | 0.77 | 0.77 | 0.52 | 0.78 | 0.33 | 0.72 | 0.74 |
| Di | | 0.466 | | 0.206 | 0.503 | | | 0.283 |
| Hd | | 0.143 | | 0.187 | 0.147 | | | 0.107 |
| Jd | | 0.252 | | 0.454 | 0.202 | | | 0.447 |
| CaTs | | 0.137 | | 0.021 | 0.111 | | | 0.054 |
| Es | | | | 0.134 | | | | 0.080 |

Glass—silicate glass, Cpx—clinopyroxene, Rec—reconstructed composition of melt inclusion (50% Cpx + 50% Glass), Omp—omphacite.

## 4. Discussion

We have interpreted the two-phase inclusions—consisting of glass and hopper crystals—in our diamond samples as melt inclusions. Hopper crystals imply rapid cooling rates, likely related to the uplift of crystals in a kimberlite melt. The glasses in the inclusions have high $SiO_2$, $Al_2O_3$, $Na_2O$ and

$K_2O$ contents. They are compositionally similar to the inclusions described by Novgorodov et al. [41] in diamonds from the Mir kimberlite pipe. At the same time, K strongly prevails over Na in all of the inclusions in the Mir diamonds, while the analysis of the inclusions in our study revealed higher Na contents (Table 1). The melt inclusions from the Mir diamonds and from our placer deposit diamonds are associated with inclusions from eclogite paragenesis (omphacite and coesite). The melt inclusions occur close to the cores of the crystals. The central inclusion in the diamond from the Mir pipe, described in [40], coexists with inclusions of eclogite paragenesis, represented by omphacite and garnet. A combined inclusion of an omphacite and K-rich silicate melt was previously found in a microdiamond from the Sytykanskaya pipe [50]. Thus, all previously known silicate-melt inclusions, as well as those described herein, coexist with the minerals of eclogite paragenesis. Melt inclusions with bulk compositions close to that of olivine have also been identified in at least one diamond [51].

The newly-formed clinopyroxenes in the studied inclusions differ from the diamond-hosted omphacites, the former having lower jadeite and higher chermakite components (Table 1). These features suggest a decrease in pressure during the formation of the clinopyroxenes [49]. This might have occurred when the diamonds were being transported by the kimberlite. The reconstructed parental melts appear to be trachyandesite-basalt. Despite the varying Mg# in omphacite inclusions co-existing in the same diamond, the melts reconstructed from the inclusions are almost identical. The clinopyroxene from Hi-51 has a Mg# similar to that of the reconstructed melt (73%), while the Mg# of the clinopyroxene from Hi-39 diamond is markedly lower than that of the reconstructed melt (Table 1). Eclogite and metabasalt melting experiments have demonstrated that the Mg# in the melt can be significantly lower than that in a co-existing clinopyroxene [49]. As these experiments have also shown, the Mg# of melts generated by the partial melting of basalts never exceeds 0.45 [48,49]. Therefore, we can assume that the inclusion-melt, formed during the melting of eclogite, changed in composition through its interaction with ultrabasic mantle lithologies. The reconstructed melts of the studied composite inclusions yielded compositions similar to those of low-$SiO_2$ adakites (Figure 5), which are also characterised by high Mg# (73–76%). According to Martin et al. [52], the primary source of high-$SiO_2$ adakites is subducted oceanic crust. The low-$SiO_2$ adakites in our samples were likely produced by the melting of a metasomatised mantle. The source of melt/fluid metasomatising the mantle can be felsic melts from subducted slab [53].

The different compositions of the omphacite and clinopyroxene in the two-phase inclusions in our diamonds suggest that the omphacite was protogenetic. This observation might also imply that the interaction between a modified mantle-wedge melt and an eclogite substrate resulted in diamond formation. Previous studies of diamondiferous eclogites have testified to the metasomatic origin of most diamonds, and have suggested their multi-stage formation [11,17,54–58]. Chloride-carbonate microinclusions have been found in fibrous diamonds in an eclogite xenolith from the Udachnaya kimberlite pipe [21]. However, studies have revealed that fibrous diamonds from kimberlite pipes of the Yakutian diamondiferous province contain HDFs with higher carbonate and silicate contents [26,29,30,59].

Sokol and Pal'yanov [60] studied the crystallisation of diamonds in the $SiO_2$–$H_2O$–C, $Mg_2SiO_4$–$H_2O$–C and $H_2O$–C systems. They showed that the most favourable conditions for diamond crystallisation occur in a water-rich fluid phase containing small amounts of a silicate solute. The degree of graphite to diamond transformation was found to decrease with a decrease in water content in the studied systems. Recent experimental works on diamond growth in hydrous $SiO_2$–carbonate-rich fluids [61,62] have confirmed the hypothesis that fibrous polycrystalline and monocrystalline diamonds grow in multi-component water-silicate melts that are rich in carbonate, and in supercritical fluids rich in carbon and water. Hydrous silicate melts and aqueous fluids can be trapped as inclusions during crystal growth, as experiments have demonstrated. Depending on the experimental conditions, diamonds can contain inclusions consisting of aqueous fluids, silicate glass or mineral grains. In the upper mantle, diamonds can grow from either supercritical fluids/melts, or from aqueous fluids equilibrated with silicate melts [61,62]. Experiments have shown that diamonds can grow in the presence of two

fluids—an aqueous fluid and a hydrous silicate melt. The source of the C for diamond growth could be carbonate ($CO_3^{2-}$) dissolved in a melt or $CO_2$ in an aqueous fluid. Therefore, the experimental data imply that water is a key component in diamond growth. Until recently, fluid inclusions were found mainly in fibrous and coated diamonds [23]. Inclusions of carbonate-bearing HDFs have been found in the twin of monocrystallime diamonds [39]. Our diamonds possibly grew from two fluids—an aqueous fluid and a silicate melt. The absence of an aqueous fluid phase in the inclusions could be explained by their depressurisation after exposure of the inclusions during polishing.

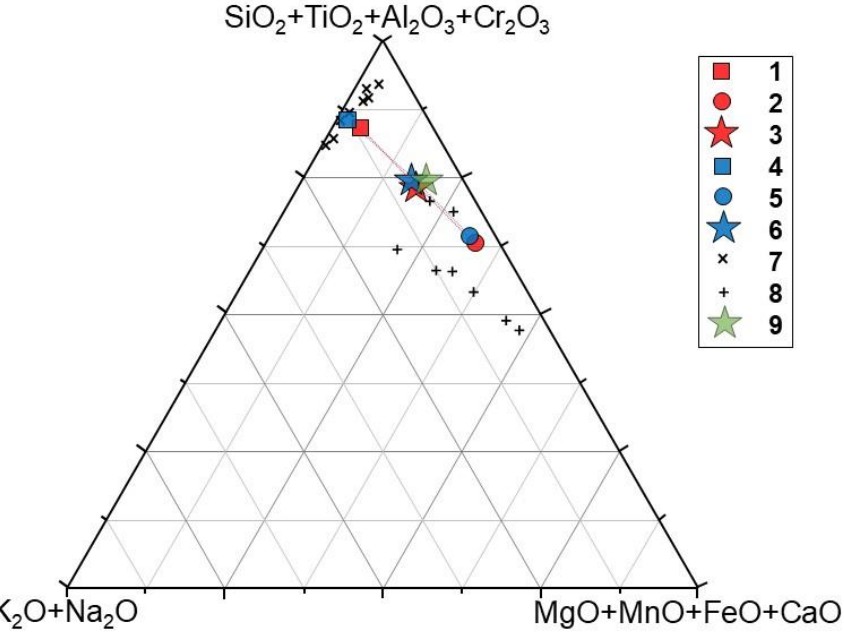

**Figure 5.** The compositional features of combined Glass + Cpx inclusions and reconstructed melts in diamonds Hi-51 (1—glass; 2—Cpx; 3—reconstructed melt) and Hi-39 (4—glass; 5—Cpx; 6—reconstructed melt). Melt inclusions in diamonds from Yakutian kimberlites (7—data from [41,42], 8—data from [51]) and 9—composition of low-Si adakites [53] are shown for comparison.

Estimates [63] have indicated that, in hot subduction zones, the subducting plate dehydrates at a depth of about 100 km. At a depth of <80 km, phengite decomposes. At depths of greater than 200 km in cold subduction zones, metagabbro may still contain water. Taking into account the Archean age of the eclogites associated with the Udachnaya kimberlite pipe [64], and the markedly higher temperatures of the upper mantle during that period [65,66], a complete dehydration of subducted eclogites at depths corresponding to the diamond stability field might be expected. However, the diamonds in many of the eclogites associated with the Udachnaya kimberlite pipe have a metasomatic nature [21,58]. Therefore, we propose that an external source of fluids caused melting of the host eclogite and subsequent diamond crystallisation. Oceanic lithosphere subducted in the Proterozoic (~2 Ga), during the amalgamation of the Siberian Craton [67], could have been the main source of such fluids. Diamonds from eclogite paragenesis strongly prevail (>85%) in the placers of the northeastern Siberian Craton [43]. The carbon isotope compositions of these diamonds, combined with the rare element compositions of their mineral inclusions, testify to diamond formation in the compositionally varying substrates of deeply subducted oceanic crust [17,21,42,43,68]. The data from this study provides additional evidence for the melting of subducted oceanic crust causing the generation of the diamonds found in the placers of the northeastern Siberian Craton.

**Supplementary Materials:** The following are available online at http://www.mdpi.com/2075-163X/9/7/412/s1, Table S1. FTIR data of studied diamonds.

**Author Contributions:** V.S. studied samples by EMP and EDS; A.R. studied inclusions by Raman spectroscopy; D.Z. conducted the cathodoluminescence experiments and analyzed the data; V.K. studied samples by optical microscopy, polished the plates, and analysed them via FTIR; V.S., D.Z. and A.R. analysed the data and discussed the results; V.S. wrote the paper.

**Funding:** Work is done on state assignment of IGM SB RAS. The research was supported in part by the RFBR (project No. 18-05-70014).

**Acknowledgments:** Critical reviews by three anonymous reviewers helped to improve manuscript considerably, for whom we are grateful.

**Conflicts of Interest:** The authors declare no conflict of interest.

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
