# Peer review of "Silicate Melt Inclusions in Diamonds of Eclogite Paragenesis from Placers on the Northeastern Siberian Craton"

_minerals, doi:10.3390/min9070412_

Round 1
Reviewer 1 Report
The manuscript studied the melt inclusion in placer diamond. These inclusions contain interesting hopper textured Cpx and glass. The original melt composition is reconstructed and the original of fluid is proposed to be subduction related metasomatism in deep mantle. These observation is interesting for the research of diamond formation. The manuscript is generally well written. Only a few points need to be considered. I would suggest a minor revision.
The introduction part could be shorter, and better organized, the current version is a bit too long.
Would it better to have a geological map, or geographic map to show the location where the diamond is collected ? You have a rather small number of figures, why not add a geological or even geographic figure ?
It is assumed the placer diamond is from the udachnaya kimberlite ? The geographic location of the placer may help the reader understand this assumption. Not everyone know the geographic relation of the placer and udachnaya kimberlite
Line 60 'show a decrease in jadeite component and increase in chermakite component'. it is better to say ‘ lower jadeite component and higher chermakite component’. In addition, the estimation of original melt composition is based on an assumption that Cpx and glass each has 50%, however, this is rather simplified. Please make a better justification how you make this assumption. How much uncertainty on the reconstructed melt composition can be produced from the uncertainty of this percentage estimation ? Would it be possible that the difference of composition comparing with omphacite is due to this uncertainty ? I understand this is a rather difficult thing, but you should make a better justification.
The current melt inclusion show hopper texture, interpreted as fast cooling during the transportation of kimberlite. However, the diamond may have formed in a rather long history, in multiple fluid processes. How to link this melt composition to the metasomatic fluid should be better justified.
Some typing errors, line 48, 50, extra space, many extra spaces, please go through all the manuscript to check this simple error.
line 56 – 64, this part is discussing diamond in metamorphic rocks, seems not really related to the topic of this manuscript on diamond from subcontinental lithospheric mantle.
line 65, don’ cover, reword this
Line 73, please define ‘FTIR’ when it appears the first time.
Table 2 should be better presented, for example, SiO2 should be SiO2
The format of reference list is not uniform, please check all the references and make the format consistent with the requirement of this journal.
Line 141, figure 1 should be changed to Figure 4
Author Response
The introduction part could be shorter, and better organized, the current version is a bit too long.
Would it better to have a geological map, or geographic map to show the location where the diamond is collected ? You have a rather small number of figures, why not add a geological or even geographic figure ?
It is assumed the placer diamond is from the udachnaya kimberlite ? The geographic location of the placer may help the reader understand this assumption. Not everyone know the geographic relation of the placer and udachnaya kimberlite
We have modified the Introduction. We add the sketch map to display the location and geological setting of the placers.
Line 60 'show a decrease in jadeite component and increase in chermakite component'. it is better to say ‘ lower jadeite component and higher chermakite component’.
Corrected.
In addition, the estimation of original melt composition is based on an assumption that Cpx and glass each has 50%, however, this is rather simplified. Please make a better justification how you make this assumption. How much uncertainty on the reconstructed melt composition can be produced from the uncertainty of this percentage estimation ?
We have roughly estimated the Cpx and glass near 50% of each from areal distribution on BSE images. We understand that this is very rough estimations but it is only way to do this.
Would it be possible that the difference of composition comparing with omphacite is due to this uncertainty ? I understand this is a rather difficult thing, but you should make a better justification.
It is not possible because glass have K content up to 5.6 wt%. At the same time K content in separate Omph inclusion less than 0.5 wt%.
The current melt inclusion show hopper texture, interpreted as fast cooling during the transportation of kimberlite. However, the diamond may have formed in a rather long history, in multiple fluid processes. How to link this melt composition to the metasomatic fluid should be better justified.
We just fix one separate event that initialized melting and diamond formation.
line 56 – 64, this part is discussing diamond in metamorphic rocks, seems not really related to the topic of this manuscript on diamond from subcontinental lithospheric mantle.
We have removed the metamorphic diamonds from Introduction and Discussion.
Reviewer 2 Report
The manuscript outlines two diamonds containing a total of two similar silicate melt inclusions and offers a geological interpretation involving slab-derived melt to explain their formation. These two diamonds are from the Kholomolokh placer in the northeastern region of the Siberian Platform. One sample contains one omphacite inclusion and one melt inclusion consisting of glass and clinopyroxene. The other sample contains two omphacite inclusions and one melt inclusion, again consisting of glass and clinopyroxene. These apparent melt inclusions are interesting and offer the potential for an important case study for diamond geology. However, the present manuscript falls short of providing a complete characterization of these melt inclusions. Unfortunately, the discussion and conclusions are a big stretch beyond the limited. While the samples and the initial data appear to be very promising, some serious work is needed before this manuscript can be made into a valuable scientific contribution.
It is not clear how the inclusions were selected and isolated in the first place. The use of microprobe and SEM requires the inclusions to be exposed so the diamond must have been polished down to intersect the inclusions. This should be described, to differentiate whether these specific inclusions were chosen somehow and the surface was polished down to intersect these inclusions, or if these inclusions were exposed by chance on a random polished surface. Do the diamonds contain any additional inclusions? Where are the inclusions with respect to the CL images in Fig 1? Why are images of only 2 of the inclusions shown, rather than showing images of all 5 studied inclusions?
For the composition of the melt inclusions, this is where the most serious problem lies. The data shown in Table 2 appear incomplete or of inadequate quality to form the basis of this entire study. It is stated in line 131 that EDS data were of poor quality (low totals) and the measurements were simply renormalized to 100%. This low total data should be reported to allow readers to assess the data quality, even if there is an additional step to renormalize the values for subsequent interpretation. It is not clear what data has been renormalized. Only one measurement shows a 100% total, so does that mean the other measurements were acceptable? The data are also called into question by the use of EDS rather than microprobe, without explicitly describing the use of a standard for calibration. These EDS measurements may therefore only be qualitative. It is not clear to the reader. Looking again at Table 2, why wasn’t FeO and MgO measured in the glass of Hi39? What is the column labeled “Inc”?
The discussion stemming from these two melt inclusions is not really justified. There is not enough evidence to make a convincing argument that this melt is derived from a subducting slab (Line 170-174). The assertion that the low totals in the glass might be caused by water (line 204) needs to be substantiated or removed. A serious omission from the discussion is that of the 2 generations of clinopyroxene that have been described from many diamond bearing eclogite xenoliths. The so-called “spongy” textured clinopyroxene is often described as a byproduct of interaction with fluids or melts associated with diamond growth. How does this modified clinopyroxene compare to the clinopyroxene-dominated melt inclusions in these diamonds?
Additional comments:
The grammar and writing style require editing throughout. It may be worth seeking help from a native English speaker.
Line 70: Describe how the polishing was done to expose the inclusions. Were these inclusions specifically selected for exposure?
Line 92: what lens was used? Where the inclusions examined before or after being exposed?
Line 137: Why assume 50%? Or do you mean this is an estimation of the relative volumes based on the exposed cross section?
Line 144: Why would the cpx crystal form at the time of kimberlite eruption? Do you propose that the melt inclusion remained molten from the time of time growth up until the time of kimberlite eruption? Do you have an idea of the melt liquidus?
Line 155: Not all silicate melt inclusions in diamonds are eclogitic in nature. Peridotitic melt inclusions have been identified in at least one diamond, with a bulk composition close to that of olivine (see Smith et al 2017 EPSL, https://doi.org/10.1016/j.epsl.2014.02.033)
Author Response
It is not clear how the inclusions were selected and isolated in the first place. The use of microprobe and SEM requires the inclusions to be exposed so the diamond must have been polished down to intersect the inclusions. This should be described, to differentiate whether these specific inclusions were chosen somehow and the surface was polished down to intersect these inclusions, or if these inclusions were exposed by chance on a random polished surface. Do the diamonds contain any additional inclusions? Where are the inclusions with respect to the CL images in Fig 1? Why are images of only 2 of the inclusions shown, rather than showing images of all 5 studied inclusions?
The two diamonds were presented by elongated rounded crystals with visible inclusions (We add the photo of crystals to Fig. 2). The surface features of both crystals reveal their intensive resorption. They were polished to expose inclusions from both sides along two parallel dodecahedral planes.
For the composition of the melt inclusions, this is where the most serious problem lies. The data shown in Table 2 appear incomplete or of inadequate quality to form the basis of this entire study. It is stated in line 131 that EDS data were of poor quality (low totals) and the measurements were simply renormalized to 100%. This low total data should be reported to allow readers to assess the data quality, even if there is an additional step to renormalize the values for subsequent interpretation. It is not clear what data has been renormalized. Only one measurement shows a 100% total, so does that mean the other measurements were acceptable? The data are also called into question by the use of EDS rather than microprobe, without explicitly describing the use of a standard for calibration. These EDS measurements may therefore only be qualitative. It is not clear to the reader. Looking again at Table 2, why wasn’t FeO and MgO measured in the glass of Hi39? What is the column labeled “Inc”?
We have add the description of analytical procedure for EDS measurements. The full protocol of EDS measurements is provided in [Lavrentiev et al., 2015 RGG]. The table 2 has been modified. Mg and Fe contents in glass phase in Hi-39 was bellow the detection limit.
The discussion stemming from these two melt inclusions is not really justified. There is not enough evidence to make a convincing argument that this melt is derived from a subducting slab (Line 170-174). The assertion that the low totals in the glass might be caused by water (line 204) needs to be substantiated or removed.
The assertion on water content is substantially revised. The studied diamonds are possibly growing from two fluids, aqueous fluid and silicate melt. The absence of aqueous fluid phase in the inclusions can be explained by their depressurization after exposition of inclusions during polishing.
A serious omission from the discussion is that of the 2 generations of clinopyroxene that have been described from many diamond bearing eclogite xenoliths. The so-called “spongy” textured clinopyroxene is often described as a byproduct of interaction with fluids or melts associated with diamond growth. How does this modified clinopyroxene compare to the clinopyroxene-dominated melt inclusions in these diamonds?
The so-called “spongy” textured clinopyroxene looks quite different from that we observed in studied inclusions (e.g. Taylor et al., 2000 IGR). Additionally Mg# of glasses from “spongy” textured clinopyroxene is very close to primary and newly formed CPx’s. In our study the Mg# of glass (0.33) in Hi-51 are significantly lower than in primary (0.73) and newly formed (0.82) Cpx’s.
Line 70: Describe how the polishing was done to expose the inclusions. Were these inclusions specifically selected for exposure?
No.
Line 92: what lens was used? Where the inclusions examined before or after being exposed?
10´, 20´, 50´. We have examined the inclusions after exposition.
Line 137: Why assume 50%? Or do you mean this is an estimation of the relative volumes based on the exposed cross section?
We have roughly estimated the Cpx and glass near 50% of each from areal distribution on BSE images. We understand that this is very rough estimations but it is only way to do this.
Line 144: Why would the cpx crystal form at the time of kimberlite eruption? Do you propose that the melt inclusion remained molten from the time of time growth up until the time of kimberlite eruption? Do you have an idea of the melt liquidus?
Hopper crystals imply fast cooling rates (e.g. Sunagawa, 1984).
Line 155: Not all silicate melt inclusions in diamonds are eclogitic in nature. Peridotitic melt inclusions have been identified in at least one diamond, with a bulk composition close to that of olivine (see Smith et al 2017 EPSL, https://doi.org/10.1016/j.epsl.2014.02.033)
This reference has been incorporated into the MS.
Reviewer 3 Report
On one hand there is good, interesting and important data. On the other hand, the paper must be revised substantially. I am sure you can revised it into a good paper that deserves being published.

Author Response
Major points:
1. The full data must be presented either in the paper or as supplementary material. This
should include the original EPMA analysis (and not only after normalization to 100%), the
FTIR and Raman spectra in an Excel format and good photos of the diamonds. The results
must include a description of the diamond. At the moment there is no data on its size,
shape, resorption, cracks, etc.
The original analysis is done in Table 2. Because it is not common to provide the Excel spreadsheets of FTIR and Raman spectra we don’t provide as supplementary in the paper. Short description of studied diamonds is provided in section 2. We also add the photo of these diamonds (see Fig.2).
2. The authors must explain why do they think that what they found, especially in Hi-39, is a
melt inclusion. The composition of the glass in Hi-39 is very close to alkali-feldspar
composition of about Na0.7K0.3Ca0.07Al1.1Si2.9O8. The other glass in Hi-51 carries little Fe and
Mg but again with similar alkalies:aluminum:silicon ratio. The Raman spectrum reveals wide
peaks, but they differ substantially from that of albite or orthoclase glasses (e.g., McKeown,
2005, Am. Min. 90, 1506–1517 and Matson et al., 1986, American Mineralogist, Volume 71,
pages 694-704) that do not absorb at 890 cm-1. Is it possible that Hi-39 trapped Cpx+feldspar
that lost its structure later?
Raman revealed the amorphous nature of the high-K and Na phase in combined inclusions. The composition of Cpx in these inclusions is strikingly different from separate Omph inclusions. Thus combined inclusions can not be interpreted just as trapped Cpx+feldspar.
3. In the discussion, you attribute the low EPMA total to the presence of water in the glass.
Can you substantiate that by Raman spectra at 3000-4000 cm-1? By FTIR through the
inclusion and showing absorption at that range?
Because of small size of inclusions we have not observed any water traces in FTIR spectra. Under laser the glass phase are not stable (burning) that make difficult to get relevant spectra in the range 3000-4000 cm-1.
4. The introduction should include a short description of the diamonds of the Kholomolokh
placer and their inclusions and at least one reference.
We add this.
5. In the "Methods" section it must be explained how the inclusions were exposed (by
polishing? by breaking the diamond?). Have you examined the inclusions by Raman before
they were exposed? Hove you tried to look for water by taking a FTIR spectrum through the
inclusion? If not can you do it now and look for water in the inclusion.
The diamonds were polished to expose inclusions from both sides along two parallel dodecahedral planes. The inclusions have not been studied before exposition. Because of small size of inclusions we have not observed any water traces in FTIR spectra.
6. The results should focus on the silicate melt and the reconstructed composition.
I has been taken into account and some changes made.
7. The discussion must be completely revised and the attention must shift from the minerals
to the melt composition. The last part is not clear and should be re-written (see minor
comments).
The discussion has been modified.
7. I think that you should compare your melt to those described by Novgorodov et al.,
Zedgenizov et al. (microdiamonds) and Bulanova et al. and to other melts found in fibrous
diamond. Best is plotting your compositions along with the other. Plotting would allow you
to present different mixing ratios of the silicate glass and the cpx, which you estimate, but
with large uncertainty.
We add the triangle plot where we compare our data with data from Novgorodov et al.,
Zedgenizov et al. (microdiamonds) and Bulanova et al and with composition of low-Si adakites.
Minor comments (by line number):
Line 29: What do you mean by "can content"? Was such xenolith ever found? If yes, it
contains for sure.
Corrected.
Line 34: "carbonates" not "carbonatites".
Corrected.
Line 62: Provide a reference
This section has been removed.
Line 63-64: The sentence is not clear, saying very little and it is not clear why the conclusion
that "diamond in metamorphic rocks can crystallize in the environments of different
composition" is important.
This section has been removed.
Line 67: The reader deserves a somewhat longer description of that not so well known
deposit, its diamonds and their inclusions + a reference.
We have modified the Introduction. We add the longer description on diamonds in the placers and sketch map to display the location and geological setting of the placers.
Line 71: If you polished on both sides a central plate "was" produced already, so the "could"
is not accurate. Have the inclusions were exposed immediately, or have they been studied
while still inside. Provide a description of the diamonds (either here, under "samples", or
better in the results. Where the inclusions isolated or connected by cracks to the surface?
The diamonds were polished to expose inclusions from both sides along two parallel dodecahedral planes. We have not observed visible cracks connected to the surface.
Line 81: 2 give you an estmate of the precision of your measurements, not the accuracy.
Corrected.
Line 106: The description of the CL should join a general description of the diamond that is
missing.
We add a general description of the diamond.
Line 109: Please provide a more detailed description of the inclusions What is their sizes,
shape and location on the CL images.
The location of inclusions is provided on Fig. 2. The size and shape of inclusions are evident from Fig. 3 and 4.
Line 114: Why not bring the size of the inclusions before the distance between them?
Scale bar is done on Fig. 3 and 4.
Line 115: Closer than what?
Corrected.
Line 116: But one omphacite has only 19% jadeite. Mention that and explain.
We deleted this analyses that was done by mistake.
Line 117: You should explain why you do not mention the first omphacite inclusion.
See line 116.
Line 129: Crystals do not "appear to be" they either "are" or not. With the EPMA and the
Raman you are sure that it is cpx.
Corrected.
Line 131: What was the total? Why if the cpx is lower than 100% you normalize, but when it
is the glass you attribute the difference to water?
We change the table.
Line 132: What makes the other 78%?
We change the table.
Line 133: I guess you meant "the second phase hosted in the inclusion".
Rephrased.
Line 135-136: In Hi-39 it is identical to alkali feldspar. Why trachyte?
Raman show that it is not alkali feldspar.
Line 138: "mass" not "volume" as you mix the oxide wheight% with no use of densities.
Corrected.
Line 138: What do you mean by "has to be"? Is it, or is it not?
Corrected.
Line 139: If you argue for presence of water, why do you compare the composition to
melting experiments in a dry system?
We also compare with experiments of Rapp and Watson, 1995 JP on dehydration melting of metabaslts.
Line 141: What do you see in the Raman at 3000-4000 cm-1?
See major comment 3.
Line 144: I agree with the fast cooling rates, but then why do you compare later Mg numbers
as if you have attained equilibrium?
This is just suggest that Omph inclusions was not in equilibrium with reconstructed melt.
Line 154: Why not compare the melts described by Novgorodav, Zedgenizov and this study
in a figure like a riangle with Si, Al and Na+K?
New plot (Fig. 6) is added.
Line 156: Table 2 should include the components of the pyroxenes (e.g., jadeite, diopside,
etc.).
Done.
Line 159: Why not start with a general description of their chemistry and then go to the
differences. Why not state that the main component is diopside?
See new Table 2.
Line 160: replace "decrease in" by "lower".
Corrected.
Line 161: May the differences between the hopper crystals and omphacites be related to
lack of equilibrium during their growth?
The Omp inclusions represent the matrix mineral equilibrated or not equilibrated with melt. The hopper Cpx was crystallysed from melt at cooling stage.
Line 165-167: The difference in Mg# is mainly the result of the zero FeO and MgO in the
residual melt of Hi-39. You cannot stick to ratios (e.g. mg#) when one component is missing.
At the end you lean on the reconstructed melt, but you do not know the mixing ratio of the
cpx and the glass.
We have roughly estimated the Cpx and glass near 50% of each from areal distribution on BSE images. We understand that this is very rough estimations but it is only way to do this.
Line 169: So may be the inclusions were never a homogeneous melt? Going straight to
adakites is a bit too far.
We believe it was homogeneous melt. Adakites is very similar to reconstructed melt compositions (See Fig. 6).
Line 179 and 184: Have you checked the EDS analyses for Cl and P?
Yes, but it was BDL.
Lines 179-182: I do not understand what you aim for? I think that you want to say that the
HDFs are evidence for metasomatic fluids. Why does it matter if the carry Cl or carbonates?
That is true! We just justify the previous results for comparison.
Line 182: HDF was spelled in the introduction.
Corrected.
Line 188: I must admit that here I have lost your line of argument. You inclusions are clearly
not a low-density hydrous fluid.
This part is modified.
Line 195: Which "two fluids can be trapped"? I do not understand.
We correct it.
Lines 198 and 200 repeat each other.
We correct it.
Line 199: Is temperature important only at 7 GPa? What the point in this long description of
the experiments?
Corrected.
Lines 203-204: All the former discussion was for stating that the difference to 100% is water?
See my above requests to check for water with Raman and IR rather than tell me stories
about experiments.
Changed.
Line 212: Why the fluids have to induce melting of the eclogite? Why not just reaction and
metasomatic interaction. Melting would affect the omphacites that you want to be
protogenetic. In any case, the melt does not look as if it is in equilibrium with the
omphacites.
We suggest that the melting was very local along to weak zones.
Line 213: So the omphacite are Archean and the metasomatism is at 2 Ga?
It is assumption based on the age determinations of eclogites from Yakutian kimberlites.
Line 217: What do you mean by: compositionally varying substrates of deeply subducted
oceanic crust? It is a complicated expression that does not say much.
See references.
Line 218: I do not think that "data from this study provide a new strong evidence for melting
of subducted oceanic crust". At best, the story you tell is compatible with the data.
Corrected.
Round 2
Reviewer 2 Report
The manuscript is much improved. Very nice! There is still some work left to be done.
There are two main issues remaining. First is that the discussion and conclusions stemming from these two melt inclusions is not really justified. It is fine to have a discussion of all of these ideas, but there is not enough evidence to make a convincing argument that this melt is derived from a subducting slab. The wording needs to be clearer that this is what the authors propose as the most likely scenario, but that it is not firmly constrained by the data. It is troubling that the abstract has not been revised at all.
The second issue is the writing itself. The grammar and writing style require editing throughout the manuscript, preferably with help from a native English speaker. This issue has not been addressed at all in this first revision.
If these can be revised, I think the paper will make a valuable contribution to the literature.
Additional comments:
Fig 1. What is the significance of the dashed line? Is it the Arctic Circle? It should be labeled or removed.
The assumed 50/50 composition (cpx/glass) used to reconstruct bulk inclusion composition needs to be explained. The assumption is based on visual estimation of the area occupied by each phase in the cross section exposed, seen in BSE images.
Fig 9. The Symbol labeled “9” is not described in the caption, leaving readers to identify it by process of elimination.
Author Response
Reply to reviewer 2.
There are two main issues remaining. First is that the discussion and conclusions stemming from these two melt inclusions is not really justified. It is fine to have a discussion of all of these ideas, but there is not enough evidence to make a convincing argument that this melt is derived from a subducting slab. The wording needs to be clearer that this is what the authors propose as the most likely scenario, but that it is not firmly constrained by the data.
Following Martin We propose that the melts are the products of a reaction between slab-derived melts and the lithospheric mantle. Such melts were produced through interaction between siliceous melts of eclogite and peridotite in experiments (Yaxley et al., 1998).
It is troubling that the abstract has not been revised at all.
The abstract has slightly been modified.
The second issue is the writing itself. The grammar and writing style require editing throughout the manuscript, preferably with help from a native English speaker. This issue has not been addressed at all in this first revision.
The English grammar has been checked by native English speaker from Cambridge proofreading service.
Fig 1. What is the significance of the dashed line? Is it the Arctic Circle? It should be labeled or removed.
The dashed is removed.
The assumed 50/50 composition (cpx/glass) used to reconstruct bulk inclusion composition needs to be explained. The assumption is based on visual estimation of the area occupied by each phase in the cross section exposed, seen in BSE images.
The polished surface provide only 2D image and we don’t know the real volume ratio of two phase. We roughly estimated ratio from 2D image.
Fig 5. The Symbol labeled “9” is not described in the caption, leaving readers to identify it by process of elimination.
Label 9 is added.